# Evaluation of Italian Simplified Matrix Test for Speech-Recognition Measurements in Noise

**Giuseppina Emma Puglisi** [1], **Federica di Berardino** [2,3], **Carla Montuschi** [4], **Fatma Sellami** [5], **Andrea Albera** [6], **Diego Zanetti** [2,3], **Roberto Albera** [6], **Arianna Astolfi** [1], **Birger Kollmeier** [5] and **Anna Warzybok** [5,*]

[1]  Department of Energy, Politecnico di Torino, 10124 Torino, Italy; giuseppina.puglisi@polito.it (G.E.P.); arianna.astolfi@polito.it (A.A.)

[2]  Department of Clinical Sciences and Community Health, University of Milan, 20122 Milan, Italy; federica.diberardino@gmail.com (F.d.B.); diego.zanetti.bs@gmail.com (D.Z.)

[3]  Audiology Unit, Department of Specialist Surgical Sciences, Fondazione IRCCS Ca' Granda Ospedale Maggiore Policlinico, 20122 Milan, Italy

[4]  Audiometrist Section of Otorhinolaryngology, Department of Surgery, Ospedale degli Infermi, 13875 Biella, Italy; carla.montuschi@aslbi.piemonte.it

[5]  Medical Physics and Cluster of Excellence Hearing4all, Carl von Ossietzky University of Oldenburg, D−26111 Oldenburg, Germany; fatma.sellami@uol.de (F.S.); birger.kollmeier@uol.de (B.K.)

[6]  Division of Otorhinolaryngology, Department of Surgical Sciences, University of Turin, 10124 Torino, Italy; aalbera@hotmail.com (A.A.); roberto.albera@unito.it (R.A.)

\*  Correspondence: a.warzybok@uol.de

**Abstract:** This study aimed at the evaluation of a simplified Italian matrix test (SiIMax) for speech-recognition measurements in noise for adults and children. Speech-recognition measurements with adults and children were conducted to examine the training effect and to establish reference speech-recognition thresholds of 50% (SRT50) and 80% (SRT80) correct responses. Test-list equivalency was evaluated only with adults. Twenty adults and 96 children—aged between 5 and 10 years—participated. Evaluation measurements with the adults confirmed the equivalence of the test lists, with a mean SRT50 of −8.0 dB and a standard deviation of 0.2 dB across the test lists. The test-specific slope (the average of the list-specific slopes) was 11.3%/dB, with a standard deviation of 0.6%/dB. For both adults and children, only one test list of 14 phrases needs to be presented to account for the training effect. For the adults, adaptive measurements of the SRT50 and SRT80 showed mean values of −7.0 ± 0.6 and −4.5 ± 1.1 dB, respectively. For children, a slight influence of age on the SRT was observed. The mean SRT50s were −5.6 ± 1.2, −5.8 ± 1.2 and −6.6 ± 1.3 dB for the children aged 5–6, 7–8 and 9–10 years, respectively. The corresponding SRT80s were −1.5 ± 2.7, −3.0 ± 1.7 and −3.7 ± 1.4 dB. High test–retest reliabilities of 1.0 and 1.1 dB for the SRT80 were obtained for the adults and children, respectively. This makes the test suitable for accurate and reliable speech-recognition measurements.

**Keywords:** speech audiometry; speech intelligibility in noise; speech-reception threshold; matrix sentence tests

## 1. Introduction

Everyday life is characterized by surrounding sounds with different characteristics, comprising speech, noise, music, and artificial and natural stimuli. To optimize the way a listener can communicate while being immersed in such a sound environment, accurate methods for the detection of hearing impairment and assessment of the listener's performance with hearing devices are needed. An assessment of hearing abilities representative for daily life can be performed clinically by means of speech audiometry tests in noise, especially if speech materials based on words or sentences are employed that are similar to everyday communication situations. However, reliable tests and procedures characterized by a high test–retest reliability, a steep slope of the intelligibility function, and a high test

sensitivity and specificity are needed for adults and children. Another desired aspect is the usability of the test for remote studies or internet-based self-test purposes. This work aimed at the evaluation of the simplified Italian matrix test (SiIMax) with children and adults considering the abovementioned aspects.

A couple of speech tests for the Italian language are available that are used in clinical practice. For adults, the disyllabic words proposed by Bocca and Pellegrini in 1950 [1] are still used as the reference speech test, distributed in 20 test lists of 10 phonetically balanced items each. However, this test presents some limitations, mainly related to the speech material being (partially) obsolete: many words do not adequately represent the modern daily lexicon anymore. In particular, almost 30% of the words included in these lists fall below the 100,000th rank of occurrence in the current Italian language, with a few words that are not in use anymore (e.g., giunco, fiele, gerla, and gelso). In order to avoid incorrect speech identification due to incomprehension of infrequently used words, Turrini et al. [2] chose 200 familiar and widely used disyllabic words from the Italian vocabulary and, after optimization with regard to phonemic balance, generated 10 lists of 20 words. The increase in speech items in a list compared to Bocca and Pellegrini's lists was studied to reduce the variability across the test lists at the speech-recognition threshold (SRT50) and allowed measuring speech intelligibility using an adaptive procedure [3].

Di Berardino et al. [4] compared the outcomes of speech audiometry tests in adults by using a sample of four different compact discs (CDs) commonly used in current clinical practice, containing phonemically balanced word lists from both the Bocca and Pellegrini, and Turrini speech tests. Upon a preliminary acoustic analysis, they found a significant difference between the recorded speech level and the recorded calibration signal level; furthermore, they noticed both subjective aspects (e.g., the speaker's lexical inflection) and objective aspects (e.g., the use of different speech materials) that differ across the available CDs and hence reduce the comparability across the tests.

Recently, a new speech test called the Italian Matrix sentence test (ITAMatrix) has been developed and validated with adults for accurate and reliable speech-recognition measurements in noise [5]. The Matrix sentence test consists of a 50-word base matrix (10 names, 10 verbs, 10 numerals, 10 adjectives, and 10 nouns, e.g., "Andrea eats many useful chairs"). From such a base matrix, semantically unpredictable sentences of a fixed grammatical structure are randomly generated. With a standard deviation of the SRT across the test lists of 0.2 dB and a test–retest reliability of 0.6 dB, the ITAMatrix test can be considered accurate and reliable. The matrix-type sentence test has been developed in a comparable way for at least 17 languages (see [6] for an overview) and can be used as an accurate tool for multilingual studies. However, the five-word sentences may be too long for the assessment of speech recognition with adults of reduced auditory memory span. A simplified version of the test consisting of three-word speech phrases instead of five-word sentences could be an alternative for this group of patients. The speech material of the simplified Matrix test (SiIMax) consists of seven out of the 10 numerals, adjectives, and nouns used in the complete version. That is, shorter speech phrases such as "five red boxes" are used instead of full sentences. Here, the SiIMax was evaluated with normal-hearing adult listeners to examine its reliability and provide reference values. The limited number of speech items combined with the closed-set response format makes the test also interesting for remote testing or even for self-test applications via tablets, computers, or mobile telephones where only a few response alternatives can be displayed. Even though the same holds for the original version of the ITAMatrix, the limited display size favors a smaller number of response alternatives and word positions in a sentence as provided by the SiIMax. Such implementations could be a subject of future work and contribute to the provision of easily accessible and affordable speech-recognition assessment for listeners with different hearing statuses and of different ages.

Concerning speech audiometry for children, the most commonly used test in the Italian language was proposed by Rimondini and Rossi Bartolucci [7]. It consists of 14 lists of 10 disyllabic words each and 14 lists of 10 meaningful sentences each. These tests are

available on a CD and can be presented either in quiet or in the presence of masking noise. Even though this test is often used for speech intelligibility measurements with children, the equivalence of the lists and test–retest reliability are unknown [8], which limits its diagnostic value.

Since speech audiometry is an essential part of the auditory test battery and the outcomes are used to decide about amplification with hearing aids, cochlear implantation or rehabilitation strategies, it is very crucial that the speech intelligibility assessment is performed with valid and reliable methods. The early detection of hearing loss in children, appropriate treatment and rehabilitation are crucial for the development of speech, language, reading and cognitive abilities [9]. The speech test must be a sensitive measure that objectively and accurately assesses specific aspects of children's speech perception abilities. This can be obtained only if the principles of psychometric theory are considered and applied by the development of speech materials. In addition to the psychophysical test properties, several aspects need to be considered and fulfilled for measurements of speech perception in children (see [9] for an overview). The cognitive and attentional demands should be age appropriate. Furthermore, performance should be independent of higher-level language abilities and vocabulary knowledge. The speech material must be suitable for measurements with different populations of children, e.g., with varying degrees of hearing impairment, and of different ages and developmental abilities.

The simplified version of the matrix sentence test was developed to account for the challenging aspects of speech audiometry in children while, at the same time, providing advantageous properties for certain applications in adults as well. To date, it has been evaluated for reliable speech-recognition measurements in normal-hearing children for German and Russian [10–12]. Whereas the full version of the matrix test can be used for reliable measurement in older children (from about 8 years old), its simplified version enables measurements with even younger children (of about 5 years) [13–15]. The less-reliable outcomes for primary school children than for adults when using the full version of the matrix test are likely related to the shorter auditory memory spans of children in that age range than for adults [15]. Neumann et al. [13] and Weißgerber et al. [14] showed that the simplified matrix test is a reliable tool not only for diagnostics but also for assessing the benefit from hearing aids and cochlear implants. Here, the SiIMax was evaluated for speech-recognition measurements in noise with children aged 5 to 10 years. Generally, this study aimed at evaluating the simplified Italian Matrix test (SiIMax) in adults and in children. Particularly, the aim of this study was (1) to test the equivalence of the test lists for normal-hearing listeners, (2) to obtain reference data for normal-hearing adult and child listeners, and (3) to assess the training effect and the test–retest reliability for both groups of listeners.

## 2. Materials and Methods

### 2.1. Speech Material

As introduced in the previous section, the speech material of the SiIMax consists of a subset of items from the original 50-word base matrix of the ITAMatrix test (7 numerals, 7 nouns and 7 adjectives). That is, shorter speech phrases such as "five red boxes" are used instead of full sentences. Each test list consists of 14 such phrases, and each word occurs twice in a test list. Table 1 shows the original base matrix with words (in bold and italics) selected for the simplified version. The phoneme distribution of the 21 words of the base matrix of the simplified version was compared with the phoneme distribution of the original version of the ITAMatrix test [5] and with a reference phoneme distribution of the Italian language taken from Tonelli et al. [16]. The three phoneme distributions are shown in Figure 1. Similarly to the study on the original version of the ITAMatrix test [5], singleton and geminate consonants were summarized as one phoneme class. Generally, the phoneme distribution follows the phoneme distribution of the ITAMatrix speech material and the language-specific distribution. The highest difference between the SiIMAx and the reference distribution is observed for the phonemes /e/ ($\Delta$ = 4.1%) and /$\varepsilon$/ ($\Delta$ = 3.7%).

These two phonemes are overrepresented in the SiIMax. The reason for this is related to the intrinsic structure of the speech phrases: since only female nouns in plural are used, all of them, as well as all adjectives (with the only exception of the adjective "grandi"—the Italian term for big), end with the phoneme /e/. Comparing the phoneme distributions of the ITAMatrix and SiIMax, the highest difference of Δ = 4.4% is observed for the phoneme /a/. In the speech material of the ITAMatrix, this phoneme is mainly included in the names and verbs. Since these word categories are not included in the speech material of the SiIMax, a drop in the occurrence of this phoneme can be noticed.

**Table 1.** Base matrix of ITAMatrix test [13] and selected words (italics/bold with a gray background) for the simplified ITAMatrix.

| Name | Verb | Numeral | Noun | Adjective | English Translation |
|---|---|---|---|---|---|
| Sofia | compra | *due* | scatole | *azzurre* | Sofia buys two light-blue boxes. |
| Marco | vuole | poche | *matite* | *piccole* | Marco wants a few small pencils. |
| Anna | prende | *quattro* | tazze | normali | Anna takes four normal cups. |
| Sara | dipinge | *cinque* | *pietre* | nuove | Sara paints five new stones. |
| Chiara | vede | molte | *tavole* | *belle* | Chiara sees many nice desks. |
| Maria | cerca | *sette* | *palle* | *bianche* | Maria looks for seven white balls. |
| Luca | trascina | *otto* | *macchine* | *grandi* | Luca drags eight big cars. |
| Andrea | regala | *nove* | *sedie* | utili | Andrea donates nine useful chairs. |
| Matteo | possiede | *dieci* | bottiglie | *nere* | Matteo owns ten black bottles. |
| Simone | manda | venti | *porte* | *rosse* | Simone sends twenty red doors. |

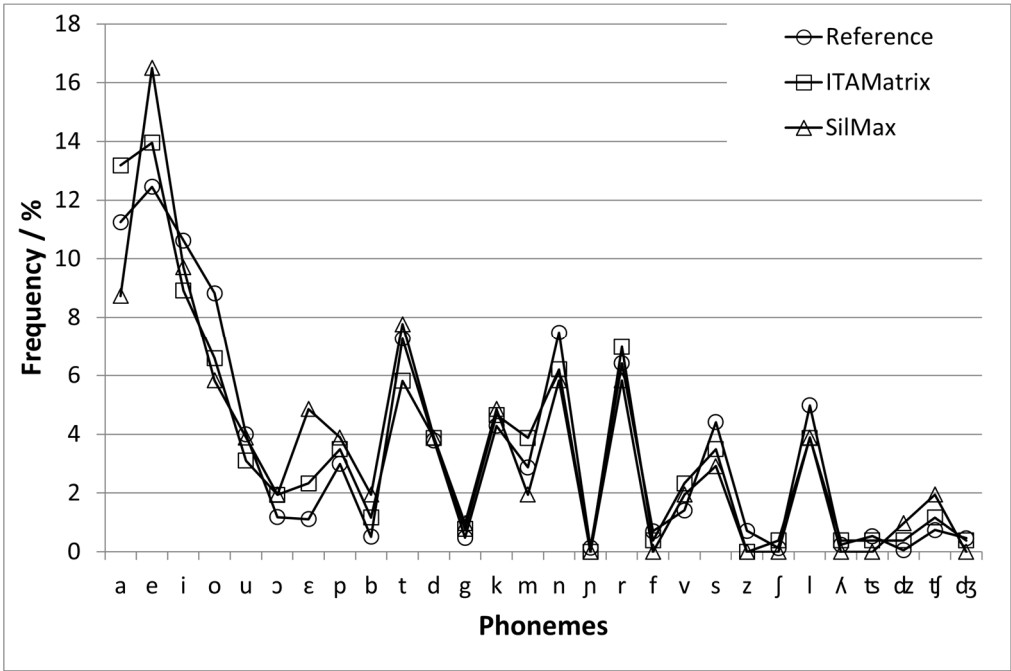

**Figure 1.** Phoneme distribution of the ITAMatrix (squares), SiIMax (triangles) and reference distribution for the Italian language (circles). The phonemes were transcribed using the symbols of the International Phonetic Alphabet.

In order to select the appropriate words for the SiIMax, the words with which school-aged children are most familiar were considered. In the class of numerals, only digits in the range of two to ten were selected, avoiding abstract words such as "pochi" (a few) or

"molti" (many). Nouns were selected based on the outcomes previously obtained with adults during the development of the original version of the ITAMatrix [5]; indeed, many listeners systematically confused certain nouns (e.g., "tazze"—cups—with "calze"—socks). Hence, keeping such words could have been misleading for very young children. From the original list of 10 adjectives, three of them were not considered in the SiIMAx, as they were the least frequently used among all. To prove the suitability of the final $3 \times 7$ matrix of words, experts in the field of pediatric audiology were involved in a preliminary validation process.

Furthermore, the number of syllables within each word group was controlled so that all the words had the same number of syllables (e.g., all the numerals were disyllabic) or a balanced number of syllables (e.g., three nouns were trisyllabic and four, disyllabics).

From the 21-word matrix, ten test lists were randomly generated, each consisting of 14 sentences such as "two new boxes". In each test list, each word occurred exactly twice. The audio .wav files were the same as the ones used in the evaluation of the original ITAMatrix test (see [5] for details). For each word, seven realizations were used to preserve the coarticulation to the subsequent word within the speech phrase; these seven realizations capture the coarticulation between the given word and all possible subsequent words.

*2.2. Evaluation Measurements*

In order to evaluate the key properties of the SiIMax test (e.g., the SRT and slope of the discrimination function, equivalence of the test lists, size of the training effect and test–retest reliability), measurements were conducted with a group of normal-hearing adult listeners and with a group of normal-hearing children aged 5 to 10 years, divided into three age subgroups. Since measurements examining the list equivalence require long evaluations with at least two different measurements per test list, this step was conducted only with adult listeners. The previous studies showed that in terms of the test list equivalence, comparable results were obtained with children and with adults [15]. Hence, it can be assumed here that the outcomes of the measurements with adults (concerning the equivalence of the test list) are valid for children as well. A shorter protocol was used with children in order to keep the evaluation time below one hour in total. Therefore, the measurements with adults will be described first, followed by the evaluation of the SiIMax with children later on. Ethical approval was obtained from the Research Ethical Committee of the Universität Oldenburg.

2.2.1. Listeners—Adults

Twenty native Italian adult listeners (12 female; 8 male) participated in the listening experiments. They were all normally hearing with a pure-tone threshold not exceeding 20 dB HL at octave audiometric frequencies of 125 to 8000 Hz. Their age ranged from 21 to 36 years, with a mean age of 24.5 years. All the listeners were informed about the purpose of the study, gave written informed consent and were paid for their participation in the listening experiments. The measurements took about 2 h per subject and were split into two measurement sessions. Each listener could take a rest whenever he/she wished.

2.2.2. Procedure—Adults

The measurements were conducted in a sound-attenuated booth in the Otolaryngology Unit, Department of Surgical Sciences, University of Turin, Italy, fulfilling the requirements of [17]. The measurement setup consisted of a notebook with an earbox "ear 3.0" sound card (Auritec, Germany) and Sennheiser HDA200 headphones (Sennheiser, Germany). The Oldenburg Measurement Applications software (HörTech GmbH, Germany) was used in order to perform the speech-recognition measurements. The setup was calibrated to dB SPL using Brüel&Kjær instruments (artificial ear type 4153, microphone 4134, preamplifier 2669 and amplifier 2610).

The measurements were performed monaurally at the listener's preferred ear. In all the conditions, the noise signal was fixed at 65 dB SPL; thus, the signal-to-noise ratio (SNR)

was determined by the speech signal level. Speech-shaped noise was used as a masker. It was the same noise as used in the development of the ITAMatrix test; i.e., it was generated from the speech material of the ITAMatrix, and its long-term power spectrum matched the average spectrum of the speech material. The noise signal was turned on and off 500 ms before and after the presentation of each sentence. The listener's task was to repeat every understood word, and the examiner marked the correctly recognized words on a display. The answers were stored using word scoring.

First, the training effect was examined. The first training list was presented at a fixed SNR of 5 dB: such a high SNR guarantees that normal-hearing listeners can understand the presented speech material and become familiar with the 21 words used in the test. The subsequent three test lists (T1, T2 and T3) were presented using a 1-up/1-down adaptive procedure with a varying step size [18] and converging to an 80% speech-recognition threshold (SRT80). As the next step, one test list was presented using the same adaptive procedure but converging to 50% of the speech-recognition rate (SRT50) in order to establish a reference value for the SRT50 and compare it with the SRT50 of the original version of the ITAMatrix test for adults, consisting of sentences with 5 words. The SRTs (SRT50 as well as SRT80) were estimated by fitting a logistic function to the measured scores using a maximum likelihood procedure that was based on all the trials in the adaptive track. This procedure presumably yielded a higher precision than estimating the SRT only from the presentation levels during the final section of each track [18]. The speech phrase order in each test list was randomized for each presentation. The indices of the test lists used in this part of the measurements were randomized across the listeners. Furthermore, the order of the speech phrases within a list was randomized in each measurement.

Subsequently, the equivalence of the test lists was evaluated. In order to obtain a list-specific recognition function with its parameters, i.e., the SRT50 and slope, speech recognition was adaptively measured at 20 and 80% speech recognition. SNRs corresponding to SRT20 and SRT80 represent a "pair of compromise" [18] that yields an efficiency-optimized simultaneous estimate of the SRT50 and slope. This pair is the best compromise between the optimum SNR for the accurate measurement of the SRT50 (the SNRs should be chosen as close as possible to the 50% recognition scores) and slope (the SNRs should result in very high and very low recognition scores). The SRT20 and SRT80 were measured for each listener with all ten lists. The order of the test-list index was randomized across the listeners. To avoid a loss of concentration and fatigue, the listeners were invited to perform eight and twelve measurements within the first and second sessions, respectively. The time between the first and the second session generally varied from 4 to 45 days. In the second measurement session, first, one measurement at a fixed SNR of 5 dB (with the same procedure as described above) was conducted for training. Subsequently, three adaptive SRT measurements with a randomly selected list index were performed, i.e., two at SRT80 (T4 and T5) and one at SRT50, in order to compare the thresholds from the first and second measurement sessions and to examine the test–retest reliability of the test. Finally, further measurements for evaluating the lists' equivalence were performed.

### 2.2.3. Listeners—Children

A total number of 96 children (51 female; 45 male) were recruited from the clinics: 44 in Turin and 52 in Milan. They were all determined to be normal hearing with pure-tone thresholds not exceeding 20 dB HL at octave audiometric frequencies of 125 to 8000 Hz. For the children measured in Turin, only children with no articulation dysfunctions and normal scores in the Fisher checklist [19] participated in the speech-recognition measurements. The articulation dysfunctions were examined by a speech therapist (one of the co-authors of this paper) using the Italian phonemic examination protocol [20]. The Fisher checklist was filled in by the parents. It aims at assessing the child's attention, auditory–visual integration, comprehension, figure–ground perception and memory. In Milan, no specific test was applied, but the parents reported normal language development and no problems with comprehension, attention, and memory for the participating children. The participants'

ages ranged between 5 and 10 years; to account for the age effect on the outcomes, the children were divided into three groups as reported in Table 2. As for the measurements carried out with adults, all the young listeners were informed about the purpose of the study, and the parents of each child agreed and signed informed consent to let their child participate in the study. All the children were invited to attend the measurements once, with each measurement session lasting about 30 min, with a break in between.

**Table 2.** Recruited listener sample for the measurements with children.

| Group | Age Range | Number of Listeners in Turin | Number of Listeners in Milan | Sex (F—Female, M—Male) |
|---|---|---|---|---|
| 1 | 5–6 | 15 | 15 | 21 F, 9 M |
| 2 | 7–8 | 13 | 18 | 17 F, 14 M |
| 3 | 9–10 | 16 | 19 | 13 F, 22 M |
| Total | 5–10 | 44 | 52 | 51 F, 45 M |

### 2.2.4. Procedure—Children

The measurements were conducted in a sound-attenuated booth fulfilling the requirements of [17] in two tertiary-care Italian University Hospitals: the Otorhinolaryngology Unit of the "A.O.U. Città della Salute e della Scienza", University of Turin, and the Audiology Unit of the "Fondazione IRCCS Ca' Granda Ospedale Maggiore Policlinico", University of Milan. As the data were collected in two different clinics, a comparison between the outcomes was also performed to evaluate the effect of the test site. The measurement setup in Turin was the same as that used in the measurements with adults. A compatible setup (a notebook with an earbox "ear 3.0" sound card and Sennheiser HDA200 headphones) was prepared and calibrated (using the same equipment) for the measurements in Milan. Again, the Oldenburg Measurement Applications software was used in order to perform the speech-recognition measurements. As in the case of the measurements with adults, the children were tested monaurally at the listener's preferred ear, with the objective of repeating aloud the understood words so that the examiner could mark the correctly recognized words on a display.

The children were first familiarized with the speech material with one test list presented at a fixed high SNR of 5 dB. The training effect was then evaluated, and, for this purpose, three subsequent measurements of the SRT80 (T1, T2 and T3) were performed using an adaptive procedure [18]. Two subsequent adaptive measurements, presented to the children in a randomized order, converged to the thresholds of 20 and 50% speech recognition. The results for the last of the three SRT80 measurements (T3), together with the SRT20, were used for the analysis of the slope of the test-specific speech-recognition function. To obtain the test-specific SRT50 and slope, a logistic function was fitted to the SRT20 and SRT80 using a maximum likelihood procedure. In all the conditions, the noise signal was fixed at 65 dB SPL and turned on and off 500 ms before and after the presentation of each speech phrase.

## 3. Results

### 3.1. Training Effect and Test–Retest Reliability—Adults

Figure 2 shows the mean SRT80 values averaged across listeners, with the corresponding standard deviations measured within the first (T1, T2 and T3) and the second (T4 and T5) measurement session, respectively. The SRT80s from the first measurement session were used to assess the training effect. The repeated-measures ANOVA conducted on the SRT80s from T1, T2 and T3 revealed no statistically significant differences across the adaptively measured thresholds ($F(1.5, 1.2) = 3.0$, $p = 0.76$). The mean SRT80 from T1, T2 and T3 was $-4.5 \pm 1.1$ dB. On average, the SRT80s measured within the first measurement session were higher by 0.9 dB than the SRT80s from the second measurement session. This difference was statistically significant ($F(1, 1.19) = 10.2$, $p = 0.003$).

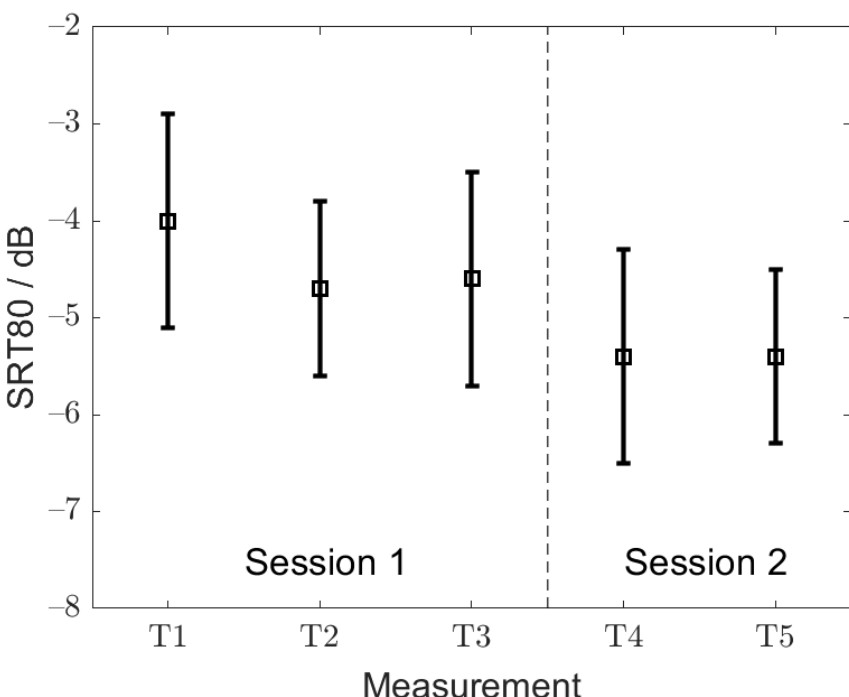

**Figure 2.** Mean SRT80s with corresponding standard deviations for training session within the 1st (T1, T2 and T3) and the 2nd measurement session (T4 and T5).

The SRT80 values from the first and second measurement sessions (T1, T2, T3, T4 and T5) were, overall, considered to assess the test–retest reliability, which is defined as the root mean square of the within-subject standard deviations of repeated SRT measures [21,22]. The test–retest reliability of the SiIMax for the SRT80 and for the adult listeners was found to be 1.0 dB. The mean SRT50s from the first and second measurement sessions were $-7.0 \pm 0.6$ and $-7.7 \pm 0.9$ dB, respectively. A comparable improvement in the SRT50 (0.7 dB) between the two measurement sessions was observed as for the SRT80. The test-retest of SRT50, calculated in the same way as for the SRT80, was 1.2 dB.

*3.2. Equivalence of Test Lists*

In order to statistically examine the equivalence of the test lists in intelligibility, repeated-measures ANOVA with "test list" as a factor was conducted separately for the SRT20 and SRT80. No statistically significant differences in the SRT80 ($F_{(9, 171)} = 1.29$, $p = 0.25$) or SRT20 ($F_{(9, 171)} = 1.35$, $p = 0.22$) across the test lists were found, confirming test-list equivalence. The mean SRT20 and SRT80, with the corresponding standard deviations, were $-10.7 \pm 1.3$ and $-5.3 \pm 1.2$ dB, respectively. To obtain list-specific parameters, i.e., the SRT50 and slope, a list-specific speech-recognition function was fitted using the logistic function based on all the data collected with the respective list (i.e., 40 data points for each list, that is, 20 for the SRT20 and 20 for the SRT80). The list-specific functions and the mean SRT and slope across all the test lists are shown in Figure 3. The mean SRT50 across the test lists was $-8.0$ dB, with a standard deviation of 0.2 dB. The test-specific slope (average of the list-specific slopes) was 11.3%/dB, with a standard deviation of 0.6%/dB. Table 3 summarizes the mean list-specific SRT20 and SRT80 (averaged across listeners) and the calculated list-specific SRT50 and slope.

To account for the variability in the SRT50 and slope across listeners, the logistic function was also fitted for each listener separately based on his/her individual data (i.e., 20 data points, that is, 10 for the SRT20 and 10 for the SRT80). The mean listener-specific SRT50 and slope with the corresponding standard deviations were $-8.0 \pm 0.6$ and $12.1 \pm 2$%/dB, respectively.

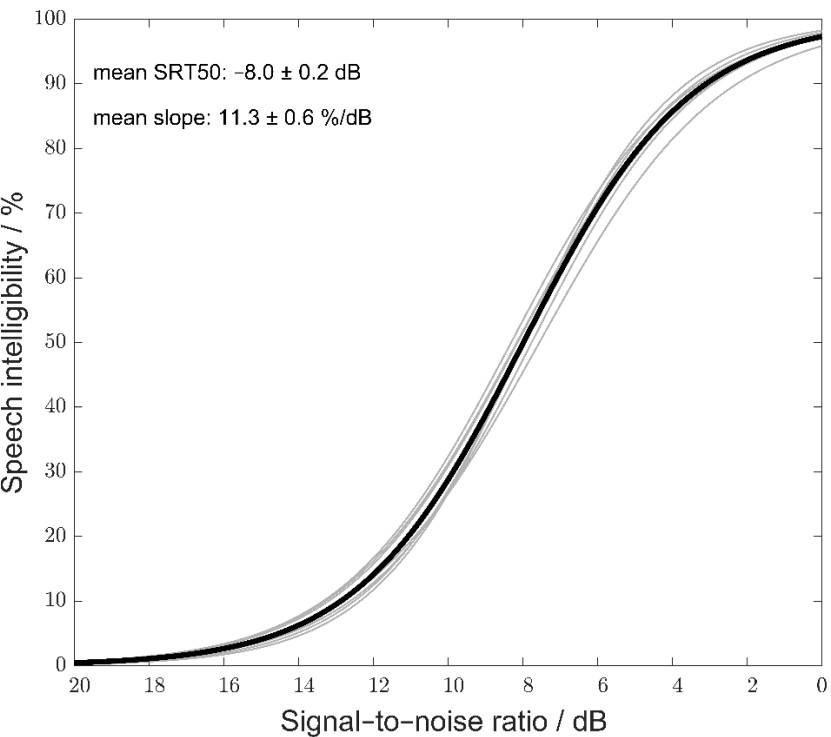

**Figure 3.** List-specific recognition functions (gray lines) and resulting average recognition function of the test (black line).

**Table 3.** Mean measured list-specific SRT20 and SRT80 averaged across listeners and SRT50 with a slope determined by fitting the logistic model function to the measured SRT20 and SRT80.

| Test-List No. | 1 | 2 | 3 | 4 | 5 | 6 | 7 | 8 | 9 | 10 | Mean | SD |
|---|---|---|---|---|---|---|---|---|---|---|---|---|
| | | | | | Measured | | | | | | | |
| SRT20 (dB) | −10.4 | −10.5 | −10.5 | −10.8 | −10.8 | −11.1 | −11.0 | −10.7 | −10.5 | −10.3 | **−10.7** | **1.3** |
| SRT80 (dB) | −4.7 | −5.1 | −5.4 | −5.4 | −5.2 | −5.6 | −5.3 | −5.6 | −5.5 | −5.6 | **−5.3** | **1.2** |
| | | | | | Calculated | | | | | | | |
| SRT50 (dB) | −7.6 | −7.8 | −8.0 | −8.0 | −8.0 | −8.3 | −8.2 | −8.1 | −8.0 | −8.0 | **−8.0** | **0.2** |
| Slope (%/dB) | 10.4 | 11.3 | 11.3 | 11.3 | 11.1 | 10.9 | 10.7 | 10.9 | 12.1 | 12.6 | **11.3** | **0.6** |

*3.3. Training Effect and Test–Retest Reliability—Children*

The first training list presented at a fixed SNR of 5 dB resulted in a median speech-recognition score of 100% for the children measured in Milan and 95% for the children measured in Turin. Considering the different age groups, the median values were 93, 98, and 99% for groups 1, 2 and 3, respectively (data pooled across the sites). For the adaptive SRT80 measurements, first, the data across the measurement sites were statistically compared, separately for each age group. Since not all the data to be compared were normally distributed, nonparametric tests on medians were applied. They revealed no statistical differences across the measurement sites in any of the age groups ($p = 0.88$, $p = 0.189$ and $p = 0.425$ for groups 1, 2 and 3, respectively). Therefore, for further analysis, the data collected in Turin and Milan were pooled together. A mixed-design ANOVA was performed on the pooled data considering the training effect as a within-subject factor and age group as a between-subject factor. Statistical differences in the SRT80s were found between the age groups ($F(2, 93) = 14.3$, $p < 0.001$). The within-subject comparisons revealed that the effect of the training list was statistically not significant ($F(2, 186) = 0.24$, $p = 0.79$), which indicates that the groups did not perform significantly better during the sequence of training lists. Furthermore, there was no interaction between the age groups and the training effect ($F(4, 186) = 0.92$, $p = 0.46$). Sidak's corrected post hoc test showed that group

2 (7–8 y.o.) and group 3 (9–10 y.o.) did not significantly differ ($p = 0.26$), but there was a significant difference between groups 1 (5–6 y.o.) and 2 ($p < 0.01$), and between groups 1 and 3 ($p < 0.001$). The mean SRT80s with the corresponding standard deviations for the respective age groups and training lists are shown in Figure 4. The highest mean SRT80 of $-1.5 \pm 2.7$ dB was observed for the youngest group of children (group 1). Groups 2 and 3 showed mean SRT80s of $-3.0 \pm 1.7$ and $-3.7 \pm 1.4$ dB, respectively. The mean SRT80 across groups 2 and 3, which did not show a statistically significant difference, was $-3.4 \pm 1.6$ dB.

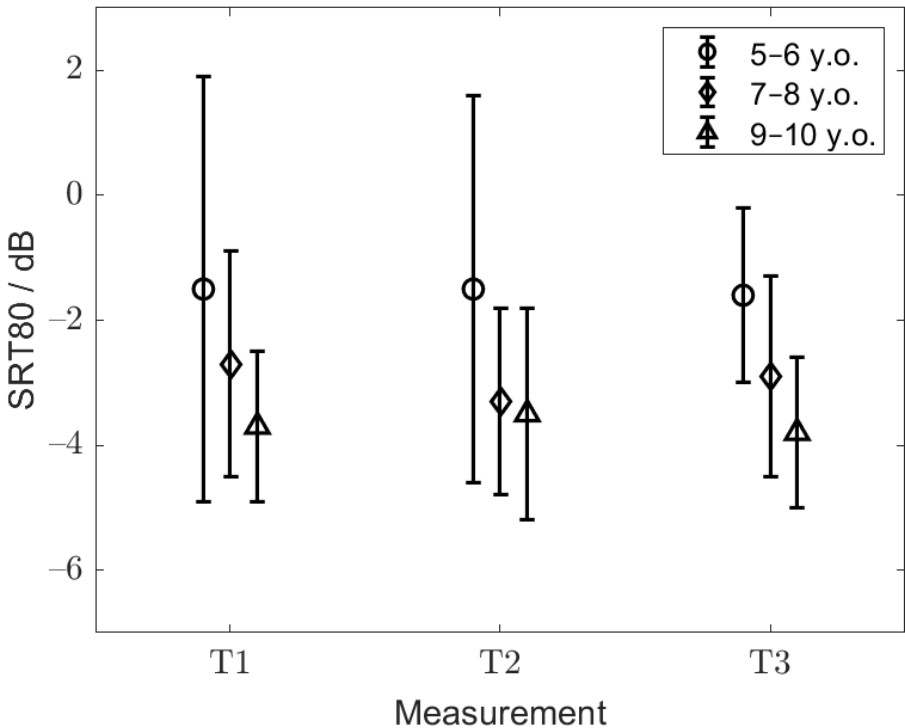

**Figure 4.** Mean SRT80 with corresponding standard deviation for the respective training list (T1–3) and age group.

Similar to the SRT80 results, no significant differences across the measurement sites were found for the SRT50s ($F(1, 94 = 0.46, p = 0.5)$). For further analyses, the data from both centers were therefore pooled together. A one-way ANOVA revealed significant differences in the SRT50s across the age groups ($F(2, 94) = 6.6, p = 0.002$). Post hoc tests with Sidak corrections showed significant differences between groups 1 and 3 ($p = 0.003$) as well as between groups 2 and 3 ($p = 0.03$). The mean SRT50s averaged across the listeners from groups 1, 2 and 3 were $-5.6 \pm 1.2$, $-5.8 \pm 1.2$ and $-6.6 \pm 1.3$ dB, respectively. Groups 1 and 2, who did not statistically differ, showed an average SRT50 of $-5.7 \pm 1.2$ dB. The test–retest reliability was assessed for each age group separately from the adaptively measured SRT80, i.e., T1, T2 and T3. It was 1.1 dB for groups 1 and 2, and 1.0 dB for group 3.

### 3.4. Test-Specific Function

In order to characterize the test-specific speech-recognition function for a particular age group, the logistic function was fitted considering the SRT20s and SRT80s of all the children within a given age group (e.g., 60 data points for group 1; 30 children x two measurements/child). The test-specific recognition function was characterized by SRT50s and slopes of $-5.0$ dB and 9.1%/dB for group 1, $-6.1$ dB and 8.9%/dB for group 2, and $-6.5$ dB and 10.6%/dB for group 3, respectively. The test-specific functions for the different age groups of children and adults are shown in Figure 5.

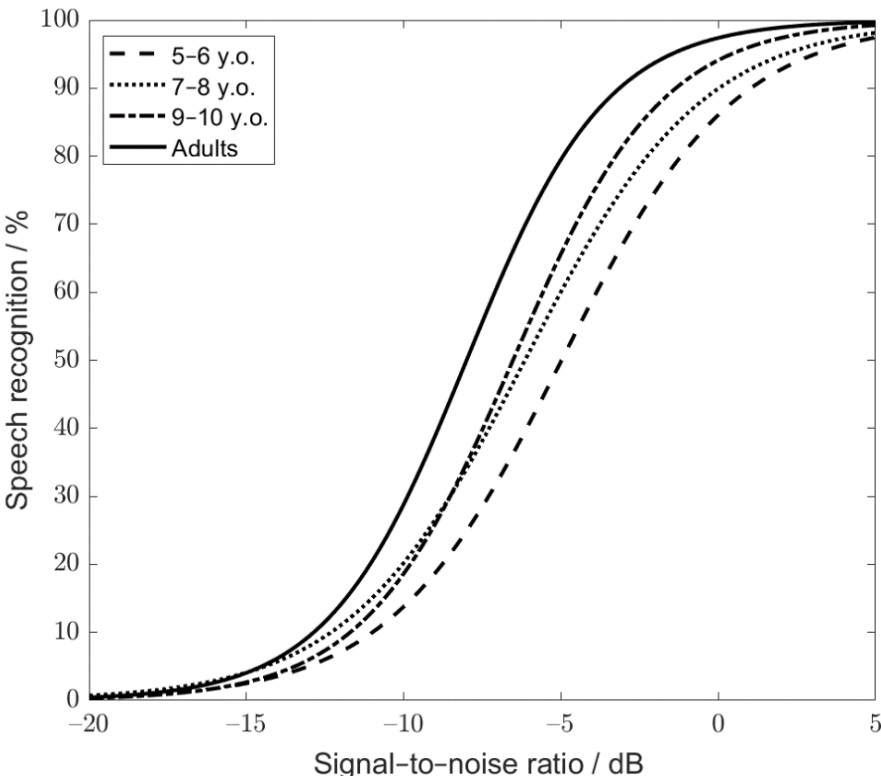

**Figure 5.** Test-specific function for 5–6 y.o. children (dashed line), 7–8 y.o. children (dash-dotted line) and 9–10 y.o. children (dotted line). The solid black line depicts the test-specific function for adults.

In order to statistically compare the parameters of the test-specific function across the age groups and account for the variability across the listeners, a logistic function was also individually fitted for each listener (based on the individual SRT20 and SRT80 values). The mean SRT50s and slopes averaged across the listeners within each independent age group were −5.0 ± 1.1 dB and 11.1 ± 3.5 dB for group 1, −6.2 ± 1.5 dB and 11.5 ± 3.3 dB for group 2, and −6.6 ± 1.0 dB and 14.0 ± 4.6 dB for group 3, respectively. The listener-specific slopes and SRT50s were analyzed by means of a two-way ANOVA across the three groups of children and the adult group. The ANOVA revealed significant differences across the age groups for the SRT50 and slope (F(3, 114) = 27.55, *p* < 0.001, and F(3, 114) = 4.16, *p* = 0.008, respectively). Post hoc tests with Sidak correction showed that the fitted SRT50 of group 1 differed significantly from those in the groups 2 (*p* = 0.001) and 3 (*p* < 0.001). No statistically significant difference in SRT50 was found between groups 2 and 3 (*p* = 0.49). All the comparisons between the SRT50s for adults and children revealed statistically significant differences (all the comparisons showed *p* < 0.001).

As far as the "Slope" factor is concerned, post hoc tests with Sidak correction indicated that the slope of speech-recognition function was statistically steeper for the listeners in group 3 than in groups 1 (*p* = 0.01) and 2 (*p* = 0.033). No difference in slope was found between groups 1 and 2 (*p* = 0.96). Additionally, the slopes obtained for the adult listeners did not differ significantly from the slopes measured with the children (*p* = 0.94 for group 1, *p* = 0.99 for group 2, and *p* = 0.29 for group 3).

## 4. Discussion

### 4.1. Adults

The measurements with the adults showed that the test lists of the SiIMax test are equivalent in terms of speech recognition; i.e., the results are not influenced by the test list employed. This indicates that the test lists can be used interchangeably. Additionally, the standard deviation of the SRT50 across the test lists (0.2 dB) was considerably smaller than

across the listeners (0.6 dB), reflecting high homogeneity of the test lists in intelligibility. The variation in the SRT across the test lists observed with the SiIMax is the same as for the ITAMatrix [5] and the German and Russian versions of the simplified Matrix test [10–12,22]. Previous studies showed that in terms of test-list equivalence, no differences were found between adults and children and that the same standard deviation of the SRT across the lists was measured [15]. Therefore, it can be assumed that the small standard deviation of the SRT50 across the test lists of the SiIMax is valid not only for adults but also for children. The mean SRT50 from the adaptive measurement as well as the variability across the listeners of the SiIMax (SRT = −7.0 ± 0.6 dB) was close to the reference data of the ITAMatrix (SRT = −6.8 ± 0.8 dB). The same observation was found for the German language [10,23]. The steepness of the test-specific slope of the SiIMax was 11.6 ± 0.6 dB%/dB. It was slightly lower than the test-specific slope of the ITAMatrix (13.3 ± 1.2%/dB). Note, however, that the slope of the SiIMax was still steeper than the slopes of typical speech tests used today in praxis [24], thus producing a higher precision of the SRT estimate for a given number of test items. Statistically, the precision for the SiIMax, with 42 words (one test list with 14 speech phrases, each of 3 words), is slightly lower than for the ITAMAtrix, which employs 100 words (one test list with 20 sentences, each of 5 sentences). This yields an approximate factor of $\sqrt{\frac{100}{42}} \times = 1.77$ higher precision (or reduction in standard error) for a single list of the ITAMAtrix in comparison to a single list of the SiIMax. Even if a single trial is considered rather than a complete test list, the ITAMatrix has a slight advantage against the SiIMax in terms of the precision gained per unit of time: within each trial, both the stimulus presentation time (which exhibits a ratio of 5/3 between the ITAMatrix and SiIMax) and the time $T_r$ required by the subject to respond and to become attentive to the next trial (which is roughly the same for both tests) have to be considered. Hence, the "effective" timing ratio is $(5 + T_r)/(3 + T_r) < (5/3)$; i.e., the time for a complete trial of the ITAMatrix is not as much larger in comparison to a complete SiIMax trial as would be expected based on the number of words presented. However, the ratio of the numbers of the collected decisions per trial is still 5/3 between the ITAMatrix and the SiIMax, which is also reflected in the slightly steeper discrimination function. Taken together, the ratio of the decisions taken per unit of time across both tests is $5/(5 + T_r):3/(3 + T_r) > 1$, indicating that the ITAMatrix yields a slightly higher test efficiency. In conclusion, the ITAMatrix is preferable if only test efficiency is considered, while the SiIMax is preferable if other test properties such as, for example, a short measurement time and low cognitive load, are more important. This may be the case for special patient groups such as children or adults of reduced auditory memory span.

Similar to for the ITAMatrix test [5], a training effect was found for the SiIMax. However, for the SiIMax, only one training list with 14 speech phrases is needed to account for the training effect, whereas in the ITAMatrix, two test training lists, each of 20 sentences, are required. Therefore, the training procedure is much shorter for the SiIMax than for the ITAMatrix. Similar observations were recorded for the German and the Russian simplified Matrix [10–12,23]. As an underlying reason, it is assumed that the shorter period to overcome the training effect for the SiIMax than for the ITAMatrix is related to the lower number of speech items (21 words instead of 50 words, respectively) and the resulting higher standard error obtained with one test list (see above), which prevents the detection of any small shift in the SRT due to training subsequent to the second test list. Only a few studies exist that have examined the training effect with different measurement procedures. In general, they indicate an effect comparable to that observed here. Primadita [25] showed that for the Indonesian Matrix test, two training lists were required even if the first training list was presented at a high, fixed SNR, resulting in very high or 100% recognition scores. In other words, the presentation of the first training list at a high SNR instead of the adaptive SRT50 measure (as was performed for the other matrix type tests) did not shorten the training effect. This provides evidence that the training effect relates to procedural learning rather than to a better acquaintance with the words available in the test lists [26]. For the simplified matrix test, Buschermöhle et al. [23] examined the training effect by

10 subsequent, adaptive SRT50 measurements. While the SRT50s of the first training list differed significantly from all the remaining measurements, the second training list showed statistically significant differences from the sixth and eighth measurements only. The magnitudes of these differences were, however, below 0.5 dB and thereby lower than the test–retest reliability of the test. Hence, the training effect occurring within the first lists in the matrix type tests seems to be comparable for different measurement procedures. A shorter measurement time is of great importance in clinical practice, in particular, with listeners of reduced auditory memory span. Hence, in these patients, the tradeoff between the measurement time and achievable precision with a given speech test calls for placing a higher weight on a shorter test such as the SiIMax even though some precision is sacrificed in comparison to the ITAMatrix. Further studies are needed to investigate the suitability and reliability of the SiIMax for measurements with the elderly population suffering from reduced auditory memory span.

Concerning the test–retest reliability, the SRT50 and SRT80 resulted in comparable reliabilities of 1.0 and 1.2 dB, respectively. These are slightly higher than the test–retest reliability of the ITAMatrix (0.6 dB for SRT50). Again, one of the possible reasons for a decreased reliability is the lower number of speech items in the SiIMax test than in the ITAMatrix (see above), where the estimated precision reduction factor of 1.77 coincides well with the observed relation of 1.0 dB/0.6 dB = 1.66. The second possible explanation is that for the original ITAMatrix test, the test–retest reliability was calculated from the adaptive measurements acquired within the same measurement session. Therefore, it did not reflect the test–retest reliability across two different testing sessions, as was the case for the SiIMax. It is also important to note that between the SRT80 measurements of the first and second measurement sessions (used for assessing the test–retest difference), eight SRT measurements were performed to examine the equivalence of the test list. This would have led to further training, which presumably accounts for the slight-but-significant improvement in the SRT80 between the two measurement sessions. Nevertheless, the test–retest reliability of the SiIMax, for both the SRT50 and SRT80, is low enough to make the test suitable for accurate speech-recognition measurements in noise.

### 4.2. Children

A short training session (only one test list needed) and the high accuracy of the SRT measurement (a test–retest reliability of about 1 dB) make the test suitable for audiological diagnostics in children. The presentation of one single test list was sufficient to make even the youngest group of tested children familiar with the speech material. A short duration is an important feature of the test since, in clinical praxis, it is important to reduce the measurement time, in order (1) to avoid/minimize fatigue and a lack of concentration of the patient and (2) to maximize the number of patients who can be diagnosed. In our multicenter experiments, the SiIMax was shown to be highly reliable. The accuracy of the SRT assessment is a prerequisite for a high sensitivity of the test, i.e., its diagnostic value in distinguishing between normal and abnormal cases. The high accuracy of the test is also important in hearing rehabilitation, including, for example, the assessment of benefits from hearing devices or monitoring of rehabilitation processes. Nevertheless, future studies are needed with hearing-impaired children in order to confirm the sensitivity of the SiIMax test.

In general, the simplified matrix type test can be used for speech-recognition measurements not only in noise but also in quiet. For German, Neumann et al. [13] showed that the simplified German matrix test is a valid audiometric test for quantifying speech perception in quiet in children from age 4. These findings were confirmed for a group of hearing-impaired children measured without and with hearing devices (hearing aids or cochlear implants).

Another important aspect related to the accuracy of SRT estimation is the slope of the speech-recognition function; i.e., a steep slope corresponds to low error in SRT estimation. With respect to the slope of the test-specific function (averaged across listener-specific

slopes), only a small difference between age groups was found, with a significantly steeper function for the oldest group. However, the magnitude of this difference (maximally 2.8%/dB between groups 1 and 3) was smaller than the minimal standard deviation within a given age group. Therefore, the observed differences across the age groups are of minor importance. Furthermore, the average slope for all the children (12.3%/dB $\pm$ 4.1) was not statistically different from the test-specific slope measured with the adults (12.1%/dB $\pm$ 2.0). For the German language, Neumann et al. [13] showed that the slope of the psychometric function for measurements in quiet was steeper than that of the commonly used German single word tests. The application of SiIMax for the assessment of speech recognition in quiet and comparison to the existing speech tests could be a subject of future studies.

Considering the SRT80 measurements, a relatively high standard deviation of 2.7 dB was observed for the youngest group of children. This was mainly due to one child who performed very poorly for the first two adaptive SRT80 measurements (T1 and T2). If the results of this single listener were excluded, the standard deviation decreased from 2.7 to 1.4 dB. The reasons for such poor performance are not clear. A normal language development for this child was reported by the parents. The youngest group required statistically higher SNRs in order to obtain 80% intelligibility than the remaining groups. The same trend was observed for the SRT50 measurements. The children in the youngest group were preschoolers or children at the very beginning of school education (the first grade of elementary school). This seems to have significantly impacted their speech intelligibility in noise performance. The children older than 7 years performed similarly, and a common reference value could be calculated for the children aged between 7 and 10 years. The mean SRT80 and SRT50 across the children from groups 2 and 3, obtained in adaptive measurements, were $-3.4 \pm 1.6$ and $-6.2 \pm 1.3$ dB, respectively. Compared to the results obtained with the adults, the children required about a 1 dB better SNR to achieve 50 or 80% speech intelligibility.

## 5. Conclusions

- The test lists of the simplified Italian matrix test deliver highly comparable intelligibility results (standard deviation of 0.2 dB across the lists at SRT50) and can be used interchangeably for repeated measurements.
- Only one training list with 14 speech items is needed to account for the training effect. This holds for adults as well as for children.
- A high test–retest reliability (approx. 1.0 dB for SRT80) makes the test suitable for accurate speech-recognition measurements in noise. This holds for adults and children. Note, however, that SRT assessment with a complete Matrix test may be more efficient; i.e., it leads to an even higher accuracy per unit of measurement time spent.
- Hence, the simplified Italian matrix test should only be preferred over the ITAMatrix test for special patient groups (e.g., children or adults with a reduced working memory capacity) or if restrictions within the measurement procedure apply (e.g., the remote testing of self-controlled test administration).

**Author Contributions:** Conceptualization, A.A. (Arianna Astolfi), B.K. and A.W.; data curation, G.E.P., F.S. and A.W.; formal analysis, G.E.P., F.S., B.K. and A.W.; funding acquisition, B.K. and A.W.; investigation, G.E.P., F.d.B., C.M., A.A. (Andrea Albera), D.Z., R.A., B.K. and A.W.; methodology, B.K. and A.W.; project administration, A.W.; resources, D.Z., R.A., A.A. (Arianna Astolfi) and B.K.; software, A.W.; supervision, F.d.B., A.A. (Arianna Astolfi), B.K. and A.W.; validation, G.E.P., F.d.B., F.S. and A.W.; visualization, F.S. and A.W.; writing—original draft, G.E.P., F.d.B., C.M., F.S., A.A. (Andrea Albera), D.Z., R.A., A.A. (Arianna Astolfi), B.K. and A.W. All authors have read and agreed to the published version of the manuscript.

**Funding:** This research was funded by Deutsche Forschungsgemeinschaft, grant number 325439187 ("Multilingual model-based rehabilitative audiology") and EXC 2177: Hearing4all (project number 390895286).

**Institutional Review Board Statement:** The study was conducted according to the guidelines of the Declaration of Helsinki, and approved by the Research Ethical Committee of the Universität Oldenburg (Drs. 75/2015).

**Informed Consent Statement:** Informed consent was obtained from all subjects involved in the study.

**Data Availability Statement:** The data presented in this study are available on request from the corresponding author. The data are not publicly available due to privacy restrictions.

**Acknowledgments:** We would like to thank all the listeners who participated in this study. The Authors are grateful to all the researchers who helped in the data collection, especially to Elena Giustolisi.

**Conflicts of Interest:** The authors declare no conflict of interest. B.K. serves as the scientific director of HörTech gGmbH (www.hoertech.de), a nonprofit organization owned in majority by Universität Oldenburg. The copyright of the speech material is held by HörTech gGmbH.

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
