# Peer review of "Evaluation of Italian Simplified Matrix Test for Speech-Recognition Measurements in Noise"

_audiolres, doi:10.3390/audiolres11010009_

Round 1

Reviewer 1 Report

See attached file.

Author Response

Dear editor, dear reviewers,

in the attachment you will find our point-by-point response to your valuable comments and modified manuscript

Kind regards,

Anna Warzybok (on behalf of all authors)

Reviewer 2 Report

Review of manuscript: Evaluation of Italian simplified matrix test for speech recognition measurements in quiet and noise

This manuscript describes the development of a new test for assessing speech intelligibility in the Italian language - the simplified Italian matrix test (SiIMax). The test is based on the original Italian Matrix Sentence Test, from which it is obtained by reducing the number of words in the test sentence. Two studies are included in the manuscript. The first one concerns the evaluation of the SiIMax with reference to young adults with normal hearing. The second study concerns the evaluation of the SiIMax with reference to children (5-10 years) with normal hearing.

This is a fairly straightforward report about validating a new clinical test and follows closely the procedures of several other matrix sentence tests created across different languages. The manuscript fits well in the aims and scopes of the Audiology Research Journal. However, I have several concerns on the aims/motivations of the manuscript and on the discussion and I believe that the manuscript requires major revisions before it can be considered for publication.

General comments

  1. The reasons for creating this test are unclear. In the Introduction it is briefly mentioned that the German version of the simplified matrix test was developed “to enable quick but accurate speech recognition measurements with children”. If this is the rationale for developing the simplified matrix test also in the Italian language, then the Introduction should be thoroughly revised and focused on the challenges of speech audiometry with children, discussing why the original ITAMatrix is not suitable for the children aged 5 to 10 years.
  2. The rationale for evaluating the test with young adults with normal hearing should be clarified. Should the simplified test instead of the full version be used with this category of listeners? At the end of the Conclusions the Authors state that the main benefit is the reduced test duration but in the manuscript there is no mention of how much the simplified version reduces this time. I agree that for clinical purposes, it is very useful that the same result is achieved in a shorter testing time but (i) will the simplified version actually reduce the testing time? (ii) the two versions (full and simplified) have a different precision, so that a shorten testing time would also result in a lower accuracy of the results.
  3. The title is misleading since the manuscript does not include data on the evaluation on the simplified test in quiet. Please remove “in quiet”.

Specific comments

Abstract:

- Line 19: the manuscript deals with adults and children with normal hearing and does not examine people with hearing impairments. Please remove the sentence.

- Line 21 (and throughout the abstract): please specify that the equivalence of the test lists was evaluated only for adults.

- Line 28: please specify SRT50 and SRT80 also for children.

- Line 30: the sentence is not supported by the data analysis. In the manuscript there is no statistical comparison between the slope of the test-specific function for adults and children. On the contrary, it is shown that the slope is significantly different between the three age groups. By looking at Figure 5 one would expect a significant difference at least between adults and children of group 1 and 2.

- Line 31: The comparability of the results between adults and children doesn’t make the test suitable for accurate speech recognition measurements. Please revise the sentence.

Introduction:

- Line 37: please revise the sentence. It is unclear how the “assessment of the benefit from a treatment” would optimize the communication in complex sound environment. Perhaps it could be replaced with the “assessment of performance of people with hearing aids”.

- Line 71: please clarify “… objective aspects that are not evenly distributed…”

- Lines 79-84: it is unclear how this paragraph relates with the other parts of the Introduction where tests for speech recognition are discussed. There are definitely other psychometric tools to assess the children performance, but especially in the areas cited by the Authors the interest lays in assessing the individual, cognitive skills and not the correct reception of the spoken message. I suggest removing the paragraph.   

- Line 94: Is this the rationale for evaluating the simplified version of the test? (please refer to General comments #1 and #2).

Materials and Methods:

- Figure 1: it can be seen that the phoneme /a/ is underrepresented in the simplified version of the test (contrary to the full version, in which it is over-represented). Maybe add a sentence in the text on this discrepancy.

- Line 157: list equivalence was evaluated only with adult listeners, thus implying that the same result would hold also for children. Could you comment on this choice?

- Line 165: how many male/female?

- Line 181: please describe (even briefly) the background noise.

- Line 191: please explain why the training effect was examined for SRT80 and not for SRT50 as in the full version of the matrix sentence tests. Please comment of the presence (or absence) of difference in the size of the training effect with reference to the chosen SRT.

- Line 187: How were the four lists used for the training selected? Were they always the same for all participants, and in the two sessions? If some of the lists were experiences twice in a session (once in the training and once for the list equivalence) couldn’t it affect the results of list equivalence?

- Line 195: please clarify why it was necessary to fit a logistic function to the outcome of the adaptive test. Isn’t the SRT80 (or SRT50) already the outcome of the measure?

- Line 218: how many male/female for each age group?

Results:

- Line 261: Please comment (in the Discussion) on the significant difference in the results of the first and the second session.

- Lines 278-279: please specify the exact p value, not only >0.05.

- Line 302: as in the following analyses the age group was considered a between-subject factor, I think that it should be considered also in this analysis. For instance, you could run three separate ANOVA, one for each age group.

- Line 350: it would be of interest to include the group of adults in the comparison, since Figure 5 reports a comparison between all age groups.

Discussion:

- Line 364: Would this result hold true also for a different population (i.e., for children?)

- Line 370: the reference data for ITAMatrix are -6.8±0.8 [Puglisi et al., 2015]

- Line 381: It isn’t correct to compare the training effect for the two versions of the test, since they were evaluated for different SRT. The paragraph starting at Line 381 bases on the assumption that the same training effect would be observed independently on the SRT but I do not think the assumption is completely true. Besides the lower number of words in the simplified version also the fact that the adaptive procedure converges to a higher SI value would facilitate the listener in learning the test material. This aspect should be acknowledged in the text, and the comparison between the training effect in the two tests revised accordingly.  

- Line 389: while I agree with the Authors on the importance of shortening the testing time for the elderly, it should be acknowledged that the results presented in the manuscript were obtained for (very) young adults. The suitability of the simplified version of the test, as well as the number of training lists, for the elderly population should be addressed in a dedicated study.

- Line 410: Would the same hold true also if the training effect was evaluated at SRT50?

- Line 426: “the accuracy of the SRT estimation… a steep slope corresponds to low error…” – might this then imply that testing (e.g. assessment of training effect) shouldn’t be performed at 80% intelligibility, where the slope is shallower?

Author Response

Dear editor, dear reviewers,

attached you will find our point-by-point responses to the reviewer's comments

Kind regards,

Anna Warzybok (on behalf of all authors)

Round 2

Reviewer 2 Report

Review of manuscript: Evaluation of Italian simplified matrix test for speech recognition measurements in noise

The Authors have addressed the majority of my original concerns in the revised manuscript, which is much improved. In particular, the Introduction has been better focused including a clearer motivation for the study. However, I still have major concerns, that I list below.

  1. Page 2, line 83: “the 5-words sentences may be too long for assessment of speech recognition with adults of reduced auditory memory span”. If the target population is “adults of reduced auditory memory span” than the Authors should revise the discussion accordingly, discussing the implication of the SiIMax development for this category of listeners, instead of the implications for the “elderly population” (Lines 462-469) or “adults with a reduced memory capacity” (Line 448).
  2. Page 2, lines 89-94: “This format of the test, due to its limited number of speech items and closed-set character is also interesting for remote testing via tablet, computer, mobile telephone or even for self-test applications.” It is not clear why the same testing could not be done using the ITAMatrix, that could be easily implemented in all the mentioned devices using the closed-test format. Also, in the experiment the SiIMax was administered in an open-test format; please revise “closed-set character”.
  3. Page 9, Line 352: The Authors did not understand my previous concern. I suggested to run separate analyses (one for each age group) to check that the SRT80 results were statistically comparable across the sites, without pooling together all children.
  4. Page 12, Lines 425-429: I suggest moving the sentences in Sec. 4.2 as they refer to the test list equivalence for children.
  5. Page 12, Lines 443-450: please rephrase the sentence, as they are unclear. For instance, which “disadvantage”? Why including a discussion on response time within each trial, when this outcome is not presented in the present manuscript (or in ref. 5)? Is the response time within each trial in ITAMatrix longer or shorter than in SiIMax? Why is SiIMax preferable for the special groups? Is this referred to the shorter format/lesser cognitive load?

6. Page 12, Lines 455-457: I am still not fully convinced that using SRT80 or SRT50 yield the same number of training lists. Has a comparison between the training effect for SRT80 and SRT50 been performed? In Ref. 6 this is not discussed and Ref. 25 could not be accessed. The Authors support their claim referring to the comparability across languages, but in all languages the training effect was examined using SRT50, and measurements types (open- or close-set format) but still in both measurements procedures SRT50 was used. Given the lack of evidences, I do not think that the claim at Line 454 is entirely justified. 

Author Response

Dear reviewer,

thank you for your valuable input and constructive comments.

We changed the text according to nearly all of them and included point-by-point responses. We provide you with an accompanying document that tracks the changes that were made to the manuscript.

Sincerly,

Anna Warzybok (on behalf of all authors)

Round 3

Reviewer 2 Report

The Authors addressed my concerns in a satisfactory manner.